# A phase 2 randomised controlled trial of mazdutide in Chinese overweight adults or adults with obesity

Linong Ji [1,15] ✉, Hongwei Jiang[2,15], Zhifeng Cheng[3,15], Wei Qiu[4,15], Lin Liao[5], Yawei Zhang[6], Xiaoli Li[7], Shuguang Pang[8], Lihui Zhang[9], Liming Chen [10], Tao Yang[11], Yan Li[12], Shen Qu [13], Jie Wen[14], Jieyu Gu[14], Huan Deng[14], Yanqi Wang[14], Li Li[14], Han Han-Zhang[14], Qingyang Ma[14] & Lei Qian [14] ✉

Mazdutide is a once-weekly glucagon-like peptide-1 (GLP-1) and glucagon receptor dual agonist. We evaluated the efficacy and safety of 24-week treatment of mazdutide up to 6 mg in Chinese overweight adults or adults with obesity, as an interim analysis of a randomised, two-part (low doses up to 6 mg and high dose of 9 mg), double-blind, placebo-controlled phase 2 trial (ClinicalTrials.gov, NCT04904913). Overweight adults (body-mass index [BMI] ≥24 kg/m²) accompanied by hyperphagia and/or at least one obesity-related comorbidity or adults with obesity (BMI ≥ 28 kg/m²) were randomly assigned (3:1:3:1:3:1) to once-weekly mazdutide 3 mg, 4.5 mg, 6 mg or matching placebo at 20 hospitals in China. The primary endpoint was the percentage change from baseline to week 24 in body weight. A total of 248 participants were randomised to mazdutide 3 mg ($n = 62$), 4.5 mg ($n = 63$), 6 mg ($n = 61$) or placebo ($n = 62$). The mean percentage changes from baseline to week 24 in body weight were −6.7% (SE 0.7) with mazdutide 3 mg, −10.4% (0.7) with 4.5 mg, −11.3% (0.7) with 6 mg and 1.0% (0.7) with placebo, with treatment difference versus placebo ranging from −7.7% to −12.3% (all $p < 0.0001$). All mazdutide doses were well tolerated and the most common adverse events included diarrhoea, nausea and upper respiratory tract infection. In summary, in Chinese overweight adults or adults with obesity, 24-week treatment with mazdutide up to 6 mg was safe and led to robust and clinically meaningful body weight reduction.

After rapidly growing prevalence over the past four decades, overweight and obesity now affect more than half of adult population in China[1]. Often clustered with hypertension, dyslipidemia, hyperglycaemia and hyperuricemia, overweight and obesity account for ever-growing non-communicable diseases-associated mortality and pose a great threat to public health[2].

Multiple lines of evidence suggest that a body weight reduction of 5–15% could significantly improve metabolic disorders and reduce the risk for type 2 diabetes, cardiovascular diseases, metabolic liver diseases and other weight-related comorbidities[3–5]. Chinese clinical practice guidelines recommend pharmacotherapy for adults with obesity or overweight adults accompanied by weight-related comorbidities inadequately controlled by 3–6 month's diet or exercise intervention[5]. However, only orlistat has been approved in China, and its clinical application is limited by modest weight loss efficacy and notable safety issues[1,5].

In 2021, the US FDA approved once-weekly semaglutide 2.4 mg for long-term body weight management[6]. In STEP 1 trial, patients with obesity receiving semaglutide 2.4 mg lost a mean body weight of 6% by week 12 and 12% by week 28, compared with a 2–3% body weight loss in

those receiving placebo[7]. Tirzepatide, a dual GLP-1 and glucose-dependent insulinotropic polypeptide receptor agonist, showed placebo-adjusted weight reduction of more than 20% after 72 weeks of treatment[8]. Other endeavour on GLP-1-based weight loss therapies came from molecules that co-activate glucagon receptors, where synergistic weight loss effect was expected given glucagon's effects to increase energy expenditure[9–11].

Mazdutide (also known as IBI362 or LY3305677), a mammalian oxyntomodulin analogue with a fatty acid side chain attached, is being developed as a once-weekly GLP-1 and glucagon receptor dual agonist for the treatment of obesity and type 2 diabetes. In a phase 1b clinical trial in Chinese overweight adults or adults with obesity, mazdutide dosed up 10 mg was well tolerated, with overall safety profiles similar to those of other GLP-1 receptor agonists and co-agonists[12,13]. At week 12, participants receiving mazdutide 6 mg and 9 mg lost a mean body weight of 6.1% and 11.7%, respectively[12,13].

Here, in a randomised, double-blind, placebo-controlled phase 2 trial, we further assess the efficacy and safety of mazdutide in Chinese overweight adults or adults with obesity. Mazdutide demonstrates robust and clinically meaningful body weight loss after 24 weeks of treatment, together with improvements on multiple cardio-metabolic risk factors. Mazdutide is well tolerated up to 6 mg over 24 weeks, with safety profile consistent with those observed in previous studies.

## Results

### Participant demographics and baseline characteristics

Between June 11, 2021 and Oct. 1, 2021, 331 people were screened and 248 were randomised (62 to mazdutide 3 mg, 63 to 4.5 mg, 61 to 6 mg and 62 to placebo). Overall, 224 participants (90.3%) completed week 24 of study treatment, while 19 (10.2%) of 186 participants with mazdutide and 5 (8.1%) of 62 with placebo discontinued the study treatment early (Fig. 1). A total of 18 participants had no data for body weight at week 24; 14 were due to early study discontinuation and 4 due to COVID-19 pandemic. All randomised participants ($n = 248$) were included in the modified intent-to-treat (mITT) and safety population. The enroled participants had a mean age of 35.5 years, a mean baseline body weight of 89.4 kg and a mean baseline body-mass index (BMI) of 31.8 kg/m². A total of 208 participants had a BMI of 28 kg/m² or greater at baseline. Baseline characteristics were essentially balanced between treatment groups (Table 1).

### Primary endpoint

In the primary analysis using analysis of covariance (ANCOVA) model, the percentage change from baseline to week 24 in body weight was dose-dependent with mazdutide. The estimated mean percentage change from baseline to week 24 in body weight were −6.7% (SE 0.7) with mazdutide 3 mg, −10.4% (0.7) with mazdutide 4.5 mg, −11.3% (0.7) with mazdutide 6 mg and 1.0% (0.7) with placebo. The estimated treatment difference versus placebo were −7.7% (95%CI: −9.5, −5.9) with mazdutide 3 mg, −11.4% (−13.2, −9.6) with mazdutide 4.5 mg and −12.3% (−14.1, −10.5) with mazdutide 6 mg ($p < 0.0001$) (Fig. 2a and Table 2). Sensitivity analyses using ANCOVA model with multiple imputation and mixed-effect model for repeated measures (MMRM) with all in-trial body weight data also supported the robust and dose-dependent body weight reduction with mazdutide at week 24. MMRM-estimated percentage reductions in body weight with mazdutide at week 24 were slightly greater than those in the primary analysis (Fig. 2b and Table S1). All mazdutide groups had significantly greater reductions in body weight than the placebo group at week 24 ($p < 0.0001$) (Fig. 2c and Table 2).

### Secondary efficacy endpoints

Significantly more participants in the mazdutide groups achieved body weight loss of 5% or more and 10% or more from baseline than those in the placebo group (Fig. 2d, Fig. S2 and Table 2). At week 24, 58.1% of participants with mazdutide 3 mg, 82.5% with mazdutide 4.5 mg and 80.3% with mazdutide 6 mg had body weight loss of 5% or more from baseline, compared with 4.8% with placebo; 19.4% of participants with mazdutide 3 mg, 49.2% with mazdutide 4.5 mg and 50.8% with mazdutide 6 mg had body weight loss of 10% or more from baseline, compared with none with placebo ($p \leq 0.0001$ for all mazdutide groups versus placebo) (Fig. 2d and Table 2). While there was no clear difference in proportion of participants achieving 5% or more and 10% or more body weight reductions from baseline to week 24 with mazdutide 4.5 mg and 6 mg, proportion of participants achieving 15% or more body weight reductions from baseline to week 24 was higher with mazdutide 6 mg (Table 2, Fig. 2d and Fig. S2).

Consistent with body weight reduction, decreases in BMI at week 24 were significant for all mazdutide doses compared with placebo (Table 2). Waist circumference reduced gradually during the treatment period. Reductions in waist circumference were significant for all mazdutide doses versus placebo at week 24, with similar reductions for mazdutide 4.5 mg and 6 mg (Fig. S3 and Table 2). Reductions in systolic blood pressure were dose-dependent with mazdutide and significant for all mazdutide doses versus placebo at week 24. Reductions in diastolic blood pressure were significant for mazdutide 4.5 mg and 6 mg versus placebo and minimal for mazdutide 3 mg at week 24 (Fig. S4 and Table 3). High-sensitivity C-reactive protein levels reduced with all mazdutide doses at week 24, more prominently with 6 mg (Fig. S5).

Slight albeit significant reductions in glycated haemoglobin $A_{1c}$ (HbA1c) and fasting plasma glucose were observed with all mazdutide doses compared with placebo at week 24 (Table 3). Of note, 35 participants had HbA1c levels of 5.7% to 6.4% at baseline. In a post-hoc analysis in this subpopulation, 6 of 9 with mazdutide 3 mg, 15 of 16 with mazdutide 4.5 mg, 10 of 10 with mazdutide 6 mg had HbA1c level of 5.6% or less at week 24, compared with 3 of 17 with placebo. Fasting insulin level reduced significantly with all mazdutide doses compared with placebo at week 24, in accordance with improvements on HOMA-IR (Table 3 and Table S2).

Reductions in total cholesterol, low density lipoprotein (LDL) cholesterol and triglycerides were significant versus placebo for most mazdutide groups at week 24, while improvement on high density lipoprotein (HDL) cholesterol was less evident. Significantly greater reductions in alanine aminotransferase (ALT) and serum uric acid were observed across all mazdutide doses compared with placebo at week 24 (Table 3).

A total of 22 participants had evaluable dual energy X-ray absorptiometry data at both baseline and week 24. With all mazdutide doses, participants lost more total fat mass than total lean mass in absolute terms (kg) (Fig. S6a). Dose-dependent decrease in the proportion of total fat mass relative to total body mass, and increase in the proportion of total lean body mass relative total body mass were observed with mazdutide (Fig. S6b). Meanwhile, visceral fat mass also decreased with mazdutide in a dose-dependent manner (Fig. S6c).

Participants had a mean IWQOL-Lite total score and physical function score of 85.8 and 88.6 at baseline, respectively. The mean change from baseline to week 24 in mean IWQOL-Lite total scores ranged from 1.7 to 5.4 across different mazdutide doses, compared with 0.5 with placebo. Changes in mean IWQOL-Lite physical function scores from baseline to week 24 were minimal.

A total of 223 participants (89.9%) attended the 12-week off-treatment follow-up visits. Body weight, BMI and waist circumference rebounded at week 36 comparing to week 24 but remained significant for all mazdutide doses versus placebo (Table S3). Reductions in systolic and diastolic blood pressure also rebounded during off-treatment period but were significant for some mazdutide doses versus placebo at week 36 (Table S3).

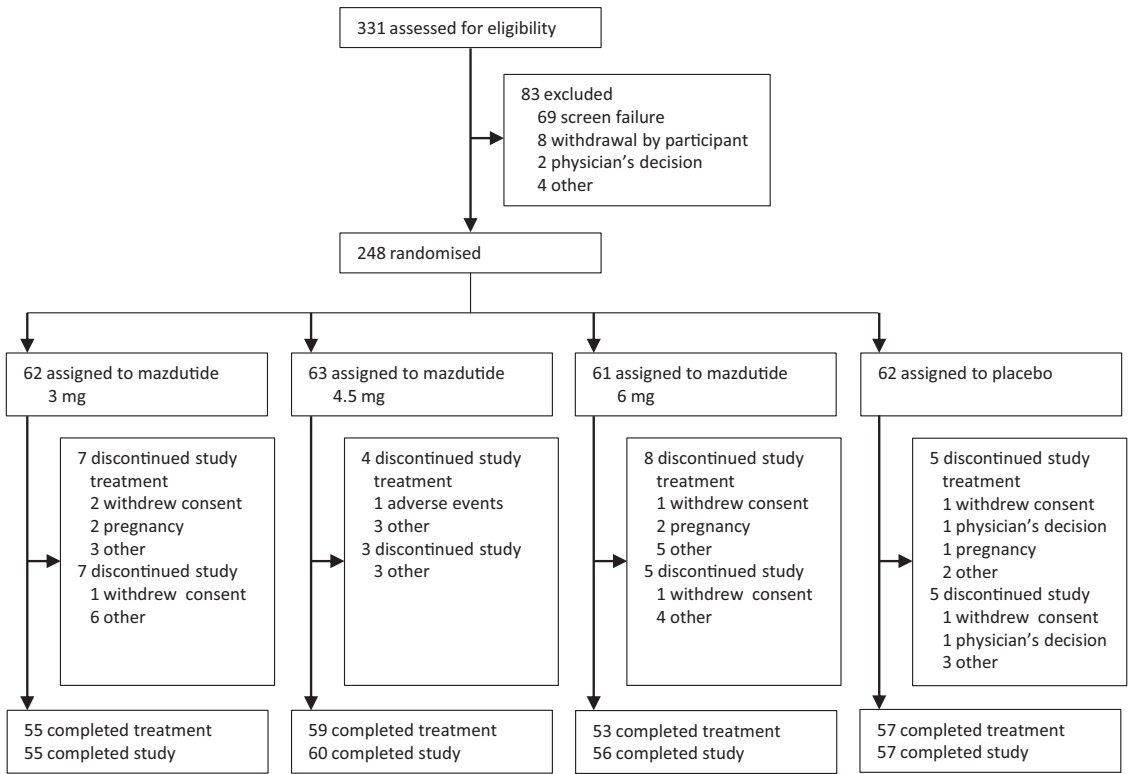

**Fig. 1 | Trial profile.** The flow diagram shows the disposition of participants screened and enrolled.

## Table 1 | Demographics and baseline characteristics

| | Mazdutide 3 mg (n = 62) | Mazdutide 4.5 mg (n = 63) | Mazdutide 6 mg (n = 61) | Placebo (n = 62) |
|---|---|---|---|---|
| Age, years | 37.2 (10.7) | 33.6 (10.0) | 35.8 (9.2) | 35.5 (7.1) |
| Sex | | | | |
| Male | 27 (43.5%) | 26 (41.3%) | 34 (55.7%) | 32 (51.6%) |
| Female | 35 (56.5%) | 37 (58.7%) | 27 (44.3%) | 30 (48.4%) |
| Race, Asian | 62 (100%) | 63 (100%) | 61 (100%) | 62 (100%) |
| BMI, kg/m$^2$ | 31.8 (3.9) | 31.8 (4.7) | 31.7 (4.0) | 32.0 (4.2) |
| Overweight (24 ≤ BMI < 28) | 9 (14.5%) | 11 (17.5%) | 11 (18.0%) | 9 (14.5%) |
| Obesity (BMI ≥ 28) | 53 (85.5%) | 52 (82.5%) | 50 (82.0%) | 53 (85.5%) |
| Body weight, kg | 89.8 (14.4) | 89.3 (15.3) | 88.5 (15.3) | 90.2 (16.4) |
| Waist circumference, cm | 104.1 (8.6) | 104.7 (11.3) | 103.9 (11.4) | 106.0 (12.0) |
| Systolic blood pressure, mmHg | 119.2 (12.0) | 116.3 (11.4) | 120.8 (9.6) | 118.5 (12.3) |
| Diastolic blood pressure, mmHg | 80.9 (7.9) | 79.4 (7.6) | 81.3 (8.5) | 80.9 (8.5) |
| HbA1c, % | 5.41 (0.27)[a] | 5.40 (0.32) | 5.34 (0.35) | 5.45 (0.35) |
| FPG, mmol/L | 5.1 (0.4) | 5.1 (0.4) | 5.1 (0.4) | 5.2 (0.6) |
| Fasting insulin, mU/L | 14.6 (10.3–22.5) | 15.1 (10.2–24.6) | 14.4 (11.1–16.9) | 16.3 (10.8–21.4) |
| Total cholesterol, mmol/L | 4.7 (4.3–5.3) | 4.7 (4.3–5.4) | 4.7 (4.4–5.2) | 4.9 (4.4–5.4) |
| LDL cholesterol, mmol/L | 3.2 (2.7–3.7) | 3.2 (2.9–3.8) | 3.2 (2.9–3.6) | 3.3 (2.9–3.8) |
| HDL cholesterol, mmol/L | 1.1 (0.9–1.2) | 1.1 (1.0–1.3) | 1.1 (1.1–1.2) | 1.1 (0.9–1.2) |
| Triglycerides, mmol/L | 1.8 (1.3–2.6) | 1.5 (1.0–2.0) | 2.0 (1.4–2.4) | 1.9 (1.2–2.8) |
| ALT, U/L | 24.5 (14.0–38.0) | 21.0 (15.0–36.0) | 24.0 (17.0–36.0) | 23.5 (15.0–36.0) |
| Serum uric acid, μmol//L | 393.1 (112.4) | 414.9 (108.6) | 422.9 (134.5) | 414.4 (111.0) |

Data are n (%), mean (SD) or median (interquartile range).

*ALT* alanine aminotransferase, *BMI* body-mass index, *FPG* fasting plasma glucose, *HbA1c* glycated haemoglobin A1c, *HDL* high-density lipoprotein, *LDL* low-density lipoprotein.

[a]n = 61.

## Safety endpoints

Treatment-emergent adverse events of any grade were reported in 95.2% of participants receiving mazdutide and 80.6% receiving placebo, mostly mild or moderate in severity. Severe events considered to be related to the study treatment by the investigators included upper abdominal pain reported in one participant with mazdutide 3 mg and diarrhoea reported in one participant with mazdutide 6 mg. Serious adverse events were reported in nine participants receiving mazdutide in no one receiving placebo, and none of the events were considered to be related to the study treatment by the investigators. Events leading

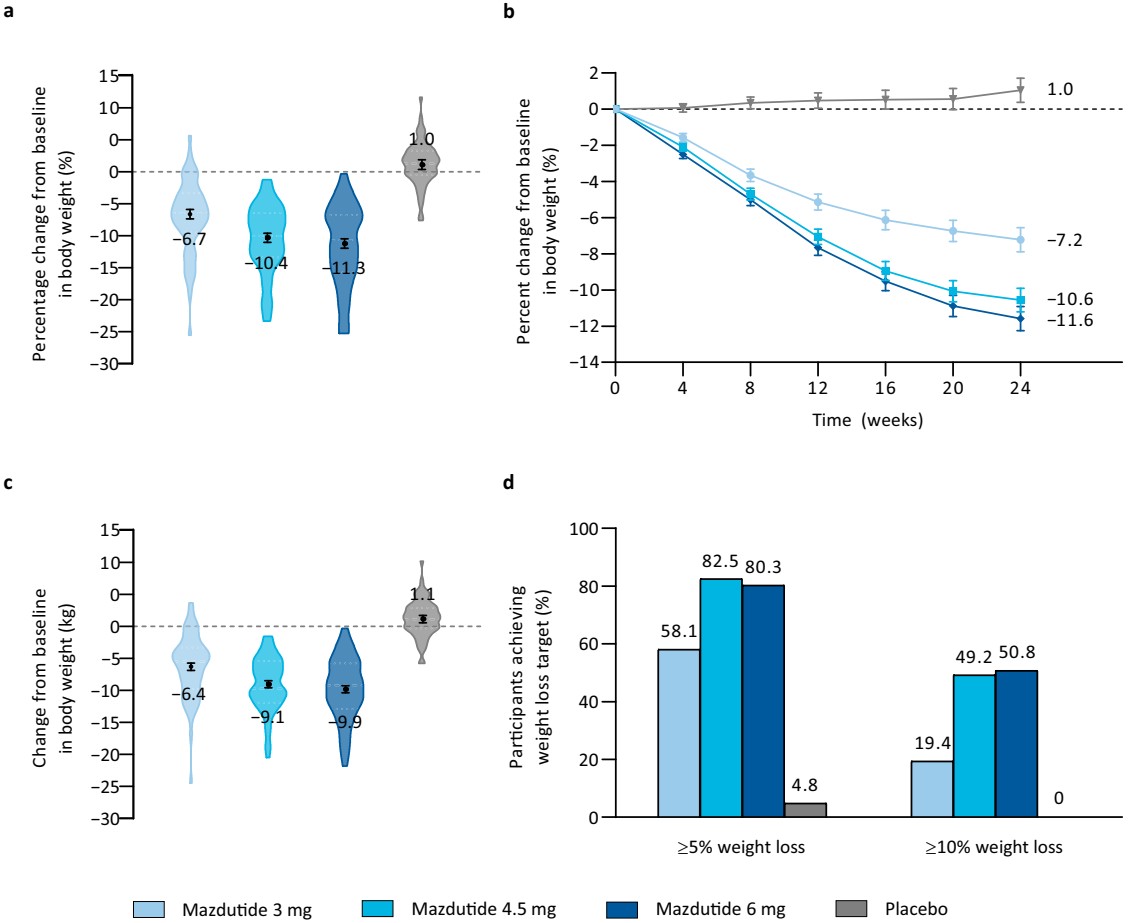

**Fig. 2 | Body weight efficacy endpoints. a** Percentage change from baseline in body weight at week 24. Symbols and error bars represent LSM and SE, from the primary analysis for the primary endpoint (ANCOVA + LOCF), mITT population. Numbers are LSM. Violin plots represent observed percentage changes from baseline in body weight for participants in the mITT population. **b** Percentage change from baseline in body weight over time. Symbols and error bars represent LSM and SE, from the sensitivity analysis for the primary endpoint (MMRM), mITT population. Numbers are LSM. **c** Change from baseline in body weight at week 24.

Symbols and error bars represent LSM and SE, from MMRM analysis, mITT population. Numbers are LSM. Violin plots represent observed changes from baseline in body weight for participants in the mITT population. **d** Proportion of participants reaching weight loss targets (≥5% and ≥10%), mITT population. Mazdutide 3 mg $n = 62$; mazdutide 4.5 mg $n = 63$; mazdutide 6 mg $n = 61$; placebo $n = 62$. ANCOVA analysis of covariate. LOCF last observation carried forward. LSM least squares mean. MMRM mixed model repeated measures. SE standard error. Source data are provided with this paper.

to treatment discontinuation were reported in only one participant with mazdutide 4.5 mg and considered unrelated to the study treatment by the investigator (Table 4). Dose reductions occurred in three participants with mazdutide 4.5 mg and seven with the mazdutide 6 mg, mostly due to gastrointestinal adverse events. Among participants who completed 24-week treatment, two participants with mazdutide 4.5 mg and five with mazdutide 6 mg were not on target doses.

Gastrointestinal symptoms (diarrhoea, nausea and vomiting) were among the most common adverse events and appeared mazdutide dose-related (Table 4). Most gastrointestinal events were mild or moderate in severity. Diarrhoea, nausea and vomiting were more frequent during dose escalation and declined gradually in the latter part of the treatment period (Fig. S7).

Only one participant with mazdutide 4.5 mg had aspartate aminotransferase (AST) exceeding three times the upper limit of normal (ULN) during the study and concurrent ALT elevation within three times the ULN at week 4 (Table S4). The elevation was transient and returned to normal for the rest of the study.

One participant with mazdutide 3 mg experienced two episodes of lipase increase above three times the ULN at week 12 and 20 and completed the study with lipase slightly above the ULN at week 36. Another participant with mazdutide 6 mg had transient lipase increase above three times the ULN at week 9. The event of marked increase in

lipase reported in the participant with mazdutide 3 mg was considered to be related to the study treatment by the investigator. No participant had amylase increase above three times the ULN during the study and no investigator-suspected pancreatitis was reported. No participant had calcitonin of 20 ng/L or higher during the study and there was no report of thyroid tumours, neoplasms or C-cell hyperplasia events (Table S4).

Cardiac disorders of clinical interest were reported in 15 participants (8.1%) with mazdutide and nine (14.5%) with placebo, with no clear association with mazdutide doses. All events were revealed by electrocardiogram, mild in severity, asymptomatic and transient. The most common cardiac disorder was sinus tachycardia, reported in seven participants (2.8%) with mazdutide and two (3.2%) with placebo (Table 4). Heart rate increase was observed in all treatment groups and returned to normal during the off-treatment follow-up period. The mean change from baseline to week 24 in heart rate were 5.82 beats per minute, 5.38 beats per minute and 8.75 beats per minute with mazdutide 3 mg, 4.5 mg and 6 mg, respectively, compared with 4.81 beats per minute with placebo (Fig. S8).

There was no observed association with mazdutide and mental health endpoints, assessed by Patient Health Questionnaire-9 (PHQ-9). No suicidal ideation or behaviour was reported based on Columbia-Suicide Severity Rating Scale (C-SSRS) assessment. Injection site

**Table 2 | Changes in body weight and waist circumference, and proportion of participants achieving weight loss targets at week 24**

| | Mazdutide 3 mg (n = 62) | | Mazdutide 4.5 mg (n = 63) | | Mazdutide 6 mg (n = 61) | | Placebo (n = 62) |
|---|---|---|---|---|---|---|---|
| | Mean | p | Mean | p | Mean | p | Mean |
| Primary endpoint - Percentage change from baseline in body weight (primary analysis – ANCOVA + LOCF) | | | | | | | |
| Percentage change, % | −6.7 (0.7) | | −10.4 (0.7) | | −11.3 (0.7) | | 1.0 (0.7) |
| ETD versus placebo | −7.7 (−9.5, −5.9) | <0.0001 | −11.4 (−13.2, −9.6) | <0.0001 | −12.3 (−14.1, −10.5) | <0.0001 | |
| Secondary endpoints | | | | | | | |
| Proportion of participants achieving | | | | | | | |
| ≥5% weight loss, n (%) | 36 (58.1) | | 52 (82.5) | | 49 (80.3) | | 3 (4.8) |
| ETD versus placebo, % | 53.2 (39.9, 66.5) | <0.0001 | 77.3 (66.4, 88.1) | <0.0001 | 75.1 (63.7, 86.5) | <0.0001 | |
| ≥10% weight loss, n (%) | 12 (19.4) | | 31 (49.2) | | 31 (50.8) | | 0 |
| ETD versus placebo, % | 19.4 (9.5, 29.2) | 0.0003 | 48.9 (36.5, 61.3) | <0.0001 | 50.1 (37.5, 62.8) | <0.0001 | |
| ≥15% weight loss, n (%) | 6 (9.7) | | 10 (15.9) | | 16 (26.2) | | 0 |
| ETD versus placebo, % | 9.7 (2.3, 17.0) | 0.0125 | 15.7 (6.7, 24.6) | 0.0013 | 25.4 (14.4, 36.4) | <0.0001 | |
| Change from baseline in body weight, kg | −6.4 (0.6) | | −9.1 (0.6) | | −9.9 (0.6) | | 1.1 (0.6) |
| ETD versus placebo | −7.4 (−9.0, −5.9) | <0.0001 | −10.2 (−11.7, −8.6) | <0.0001 | −10.9 (−12.5, −9.4) | <0.0001 | |
| Change from baseline in BMI, kg/m² | −2.3 (0.2) | | −3.3 (0.2) | | −3.6 (0.2) | | 0.4 (0.2) |
| ETD versus placebo | −2.7 (−3.2, −2.1) | <0.0001 | −3.7 (−4.2, −3.1) | <0.0001 | −4.0 (−4.6, −3.4) | <0.0001 | |
| Change from baseline in waist circumference, cm | −5.6 (0.7) | | −8.5 (0.7) | | −8.8 (0.7) | | −1.1 (0.7) |
| ETD versus placebo | −4.5 (−6.5, −2.4) | <0.0001 | −7.3 (−9.3, −5.3) | <0.0001 | −7.6 (−9.6, −5.6) | <0.0001 | |

For continuous variables, data are LSM (SE) for change and percentage change from baseline and LSM (95% CI) for ETD, from MMRM (secondary endpoints) or ANCOVA (primary endpoint) analysis, mITT population. For categorical varibles, data are n (%) and ETD (95% CI) from Mantel-Haenszel analysis, mITT population. Proportion of participants reaching weight loss targets (≥5%, ≥10% and ≥15%) was obtained by dividing the number of participants reaching respective targets at week 24 by the number of participants with baseline value and at least one non-missing post-baseline value. Participants with missing value at week 24 were treated as non-responders. All statistical tests were two-sided at a significance level of 0.05, and no adjustments were made for multiplicity. P values were nominal except for those of the primary analysis for the primary endpoint.

*ANCOVA* analysis of covariate, *ETD* estimated treatment difference, *LOCF* last observation carried forward, *LSM* least squares mean, *MMRM* mixed model repeated measures.

reaction and hypersensitivity events were infrequent, with overall incidence comparable between participants receiving mazdutide and placebo. Hypoglycaemia events were reported in five participants (2.0%) with mazdutide and one (1.6%) with placebo. All hypoglycaemia events were asymptomatic and none was graded as severe (Table 4).

Treatment-induced anti-mazdutide antibodies were detected in 22.6–32.8% of participants with different mazdutide doses and appeared to be dose-related (Table S4).

## Discussion

In this randomised, double-blind, placebo-controlled phase 2 trial, once-weekly mazdutide dosed up to 6 mg for 24 weeks achieved significant and clinically relevant body weight reduction in Chinese overweight adults or adults with obesity, together with improvements on multiple cardio-metabolic risk factors. Mazdutide was well tolerated in all doses and showed an overall favourable safety profile, similar to other GLP-1 receptor agonists and co-agonists[14]. This study provides important evidence on the efficacy and safety of GLP-1-based agents in a Chinese population and supports the potential of mazdutide as a potent GLP-1 and glucagon dual agonist for the treatment of overweight and obesity.

Since Chinese people are likely to have higher body fat content and higher cardiovascular risks than White people at given BMI levels, BMI cut-offs for overweight and obesity in China (24 kg/m² for overweight and 28 kg/m² for obesity) are lower than the WHO cut-offs[2]. While semaglutide demonstrated robust body weight loss in people of various ethnicity[7,15], clinical evidence supporting the effects of GLP-1-based agents in Chinese overweight adults or adults with obesity remain limited. In STEP 7 trial, semaglutide, in combination with diet and physical activity, achieved 44-week mean body weight reduction of 12.1% in a predominantly Chinese adults population. However, this trial excluded participants with BMI less than 27 kg/m²[16]. According to the most recent national data, 34.3% of Chinese adults were overweight based on Chinese criteria[1]. Thus, it is important to

investigate how Chinese overweight adults accompanied by weight-related comorbidities and who failed to achieve desired weight loss target using lifestyle intervention alone respond to GLP-1-based pharmacotherapy. In our study, results from 16.1% of overweight participants at baseline will provide important evidence on the applicability of mazdutide for the weight management of Chinese overweight adults.

On the other hand, 83.9% of the participants were classified as obese at baseline and 62.9% had baseline BMI of 30 kg/m² or higher. While 24-week treatment with mazdutide 4.5 mg or 6 mg may not be enough to achieve desired weight loss target for people with higher BMI, further reductions in body weight were desirable and achievable with extended treatment, as evidenced by the declining trend of mean body weight at week 24. A 48-week phase 3 trial is underway to evaluate the long-term efficacy and relative potency of mazdutide 4 mg and 6 mg in Chinese overweight adults or adults with obesity. Moreover, mazdutide 9 mg, exclusively for adults with BMI of 30 kg/m² or higher, is under evaluation in the second part of this study.

Waist circumference, with high correlation with visceral adiposity, is independently associated with cardiovascular risk[1,17]. It is reported that Chinese people have higher amounts of visceral fat than White people for the same waist circumference, underscoring the need for lower waist circumference targets[17]. In this study, 96% of participants had as central obesity based on the Chinese criteria (90 cm in men and 85 cm in women)[5]. Among these, less than one fifth with mazdutide achieved the targets at week 24, despite an overall marked reductions observed with mazdutide 4.5 mg and 6 mg. This result further implied that 24-week treatment with mazdutide may be relatively short for sufficient reduction in waist circumference and associated cardiovascular risk.

Reductions in HbA1c, fasting plasma glucose and HOMA-IR were observed with all mazdutide doses, implying potential glycaemic benefits and improvement on insulin resistance in non-diabetic Chinese overweight adults or adults with obesity. Improvements on other cardio-metabolic risk factors, including blood pressure and lipids,

**Table 3 | Changes in cardio-metabolic risk factors at week 24**

|  | Mazdutide 3 mg (n = 62) | | Mazdutide 4.5 mg (n = 63) | | Mazdutide 6 mg (n = 61) | | Placebo (n = 62) |
|---|---|---|---|---|---|---|---|
|  | Mean | p | Mean | p | Mean | p | Mean |
| Change from baseline in | | | | | | | |
| Systolic blood pressure, mmHg | −1.3 (1.3) | | −2.2 (1.2) | | −6.9 (1.2) | | 2.9 (1.2) |
| ETD versus placebo | −4.2 (−7.5, −0.9) | 0.0137 | −5.2 (−8.4, −1.9) | 0.0021 | −9.9 (−13.2, −6.6) | <0.0001 | |
| Diastolic blood pressure, mmHg | −0.6 (0.8) | | −1.7 (0.8) | | −3.6 (0.8) | | 0.9 (0.8) |
| ETD versus placebo | −1.5 (−3.7, 0.7) | 0.1876 | −2.6 (−4.7, −0.4) | 0.0222 | −4.4 (−6.7, −2.2) | 0.0001 | |
| HbA1c, % | −0.14 (0.03) | | −0.24 (0.03) | | −0.21 (0.03) | | 0.16 (0.03) |
| ETD versus placebo | −0.30 (−0.37, −0.24) | <0.0001 | −0.40 (−0.47, −0.33) | <0.0001 | −0.37 (−0.44, −0.30) | <0.0001 | |
| FPG, mmol/L | −0.2 (0.1) | | −0.4 (0.1) | | −0.4 (0.1) | | 0.1 (0.1) |
| ETD versus placebo | −0.3 (−0.5, −0.2) | 0.0003 | −0.5 (−0.6, −0.3) | <0.0001 | −0.4 (−0.6, −0.3) | <0.0001 | |
| Fasting insulin, mU/L | −5.1 (1.5) | | −7.2 (1.4) | | −6.9 (1.5) | | −0.7 (1.5) |
| ETD versus placebo | −4.4 (−8.3, −0.5) | 0.0267 | −6.5 (−10.3, −2.7) | 0.0009 | −6.2 (−10.1, −2.3) | 0.0018 | |
| Serum uric acid, µmol//L | −81.9 (10.1) | | −88.2 (9.5) | | −105.9 (9.7) | | −33.3 (10.0) |
| ETD versus placebo | −48.6 (−74.7, −22.5) | 0.0003 | −54.9 (−80.4, −29.4) | <0.0001 | −72.6 (−98.5, −46.8) | <0.0001 | |
| Percentage change from baseline in | | | | | | | |
| Total cholesterol, % | −6.2 (2.0) | | −10.5 (1.9) | | −12.2 (2.0) | | 1.3 (2.0) |
| ETD versus placebo | −7.5 (−12.8, −2.2) | 0.0057 | −11.8 (−17.0, −6.6) | <0.0001 | −13.5 (−18.8, −8.3) | <0.0001 | |
| LDL cholesterol, % | −3.7 (2.4) | | −8.8 (2.3) | | −11.1 (2.3) | | 2.5 (2.4) |
| ETD versus placebo | −6.3 (−12.5, 0.0) | 0.0504 | −11.3 (−17.4, −5.2) | 0.0004 | −13.6 (−19.8, −7.4) | <0.0001 | |
| HDL cholesterol, % | 4.3 (1.8) | | 1.2 (1.7) | | −0.1 (1.7) | | 7.4 (1.8) |
| ETD versus placebo | −3.1 (−7.7, 1.6) | 0.1919 | −6.3 (−10.8, −1.7) | 0.0072 | −7.6 (−12.2, −3.0) | 0.0014 | |
| Triglycerides, % | −26.6 (4.5) | | −36.4 (4.3) | | −34.9 (4.4) | | 0.2 (4.5) |
| ETD versus placebo | −26.8 (−38.4, −15.2) | <0.0001 | −36.5 (−48.1, −25.0) | <0.0001 | −35.1 (−46.6, −23.6) | <0.0001 | |
| ALT, % | −29.0 (5.2) | | −25.9 (5.0) | | −22.6 (5.1) | | −3.4 (5.2) |
| ETD versus placebo | −25.6 (−39.0, −12.2) | 0.0002 | −22.5 (−35.7, −9.4) | 0.0009 | −19.2 (−32.6, −5.9) | 0.0049 | |

Data are LSM (SE) for change from baseline and LSM (95% CI) for ETD versus placebo at week 24 from MMRM analysis, mITT population. All statistical tests were two-sided at a significance level of 0.05, and no adjustments were made for multiplicity. P values were nominal.

ALT alanine aminotransferase, BMI body-mass index, ETD estimated treatment difference, FPG fasting plasma glucose, HbA1c glycated haemoglobin A1c, HDL high-density lipoprotein, LDL low-density lipoprotein, LSM least squares mean, MMRM mixed model repeated measures.

were observed with all mazdutide doses, most notably with mazdutide 4.5 and 6 mg. Robust reductions in ALT and serum uric acid levels were evident with mazdutide, irrespective of doses. In this study, hyperlipidaemia (41.9%), hepatic steatosis (33.5%), hyperuricaemia (30.2%) and hypertension (16.5%) were the most common concomitant diseases at baseline. Reductions in blood pressure, lipids, ALT and serum uric acid provide comprehensive cardio-metabolic benefits to Chinese overweight adults or adults with obesity, dramatically alleviating metabolic disorders.

The most common adverse events reported with mazdutide were mainly gastrointestinal symptoms, which was consistent with the safety profiles of GLP-1-based therapies. Higher incidence of nausea, diarrhoea and vomiting were generally associated with higher doses of mazdutide. Most gastrointestinal symptoms were mild and moderate in severity, transient, and manageable with concomitant medication to alleviate symptoms or dose reductions. Importantly, the low incidence of adverse events leading to study drug discontinuation is a major strength of this study. Chronic weight management often requires long-term medication. Intolerable gastrointestinal adverse events may lead to early discontinuation and is of special concern for GLP-1 receptor agonists and co-agonists[18]. In clinical studies of semaglutide, incidence of adverse events leading to treatment discontinuation was generally low[19]. However, high rate of treatment discontinuation was reported with JNJ-64565111 (up to 32.2%) in a phase 2 study in individuals with obesity and with BI 456906 (up to 30.0%) in a phase 2 study in patients with type 2 diabetes[11,20]. The overall favourable tolerability and safety profiles of mazdutide would ensure medication compliance for optimal benefits.

Due to the cardio-stimulant effects of glucagon, more pronounced increase in heart rate is expected with GLP-1 and glucagon receptor dual agonists than GLP-1 receptor mono-agonists[21]. Indeed, a mean increase in heart rate of 5–15 beats per minute was associated with cotadutide[22], SAR425899[23], JNJ-64565111[11], NN1177[9], BI456906[24] and mazdutide[12,13]. In the multiple-ascending-dose phase 1b study of BI 456906, six participants reported cardiac or vascular adverse events that led to treatment discontinuation[24]. In the phase 1b study of mazdutide in Chinese overweight participants or participants with obesity, heart rate increases up to 15 beats per minutes was observed with mazdutide 6 mg and no further increase was observed with higher doses of 9 mg and 10 mg[12,13]. In this study, the trend of heart rate increase in the first 12 weeks was similar to that in the phase 1b study. In the latter part of the treatment period, no further increase in heart rate was observed with all mazdutide doses, with end-of-treatment heart rate increase similar to or slightly higher than that observed with placebo. In general, except for a few transient sinus tachycardia reported with mazdutide 4.5 mg, no clear association with cardiac disorders was observed with mazdutide. Nevertheless, cardiovascular benefits and risks with mazdutide should be further evaluated in a larger and longer study.

The main limitation of this study is the relatively short treatment period. As discussed above, the majority of participants with BMI greater than 30 kg/m$^2$ may need long-term pharmacotherapy to achieve desired body weight and comprehensive cardio-metabolic benefits. Moreover, due to COVID-19 pandemic-related restrictions during the conduct of this trial, participants had limited access to DEXA instruments at the hospitals. Data on exploratory endpoints of

**Table 4 | Treatment-emergent adverse events**

| | Mazdutide 3 mg (n = 62) | Mazdutide 4.5 mg (n = 63) | Mazdutide 6 mg (n = 61) | Placebo (n = 62) |
|---|---|---|---|---|
| Any TEAEs | 58 (93.5) | 60 (95.2) | 59 (96.7) | 50 (80.6) |
| Mild | 43 (69.4) | 32 (50.8) | 29 (47.5) | 41 (66.1) |
| Moderate | 14 (22.6) | 28 (44.4) | 28 (45.9) | 9 (14.5) |
| Severe | 1 (1.6) | 0 | 2 (3.3) | 0 |
| Serious TEAEs | 1 (1.6) | 4 (6.3) | 4 (6.6) | 0 |
| TEAEs leading to treatment discontinuation | 0 | 1 (1.6) | 0 | 0 |
| Adverse events occurring in at least 10% of participants in any treatment group[a] | | | | |
| Diarrhoea | 12 (19.4) | 19 (30.2) | 19 (31.1) | 9 (14.5) |
| Nausea | 13 (21.0) | 15 (23.8) | 25 (41.0) | 3 (4.8) |
| Upper respiratory tract Infection | 12 (19.4) | 11 (17.5) | 19 (31.1) | 11 (17.7) |
| Hepatic steatosis | 13 (21.0) | 10 (15.9) | 7 (11.5) | 13 (21.0) |
| Vomiting | 8 (12.9) | 13 (20.6) | 17 (27.9) | 2 (3.2) |
| Hyperuricaemia | 8 (12.9) | 13 (20.6) | 5 (8.2) | 14 (22.6) |
| Decreased appetite | 6 (9.7) | 10 (15.9) | 18 (29.5) | 5 (8.1) |
| Urinary tract infection | 6 (9.7) | 13 (20.6) | 7 (11.5) | 4 (6.5) |
| Hyperlipidaemia | 9 (14.5) | 2 (3.2) | 4 (6.6) | 9 (14.5) |
| Abdominal distension | 6 (9.7) | 7 (11.1) | 7 (11.5) | 2 (3.2) |
| Other TEAEs of clinical interest | | | | |
| Total hypoglycaemia (plasma glucose ≤3.9 mmol/L) | 1 (1.6) | 2 (3.2) | 2 (3.3) | 1 (1.6) |
| Severe hypoglycaemia | 0 | 0 | 0 | 0 |
| Injection site reaction | 3 (4.8) | 1 (1.6) | 2 (3.3) | 4 (6.5) |
| Hypersensitivity | 1 (1.6) | 1 (1.6) | 0 | 1 (1.6) |
| Cardiac disorders[a] | 2 (3.2) | 9 (14.3) | 4 (6.6) | 9 (14.5) |
| Sinus tachycardia | 1 (1.6) | 5 (7.9) | 1 (1.6) | 2 (3.2) |
| Sinus arrhythmia | 1 (1.6) | 2 (3.2) | 1 (1.6) | 4 (6.5) |
| Supraventricular extrasystoles | 2 (3.2) | 0 | 2 (3.3) | 2 (3.2) |
| Sinus bradycardia | 0 | 2 (3.2) | 0 | 2 (3.2) |
| Ventricular extrasystoles | 1 (1.6) | 1 (1.6) | 0 | 0 |
| Arrhythmia supraventricular | 0 | 1 (1.6) | 0 | 0 |
| Tachycardia | 1 (1.6) | 0 | 0 | 0 |

Data are n (%), safety population.

TEAE treatment-emergent adverse event.

[a]By the Medical Dictionary for Regulatory Activities (MedDRA, version 24.0) preferred term.

body composition measurement were limited and preliminary in this study. Lastly, all participants were Chinese, limiting the generalisability of the results to patients of other races or ethnicities.

In summary, 24-week treatment with mazdutide at doses of 3 mg, 4.5 mg and 6 mg all led to clinically meaningful and significant body weight reductions versus placebo. Compared with placebo, greater improvements on multiple cardio-metabolic risk factors were observed with mazdutide. Mazdutide dosed up to 6 mg showed favourable tolerability and safety profiles. Taken together, these data support future developments of mazdutide for long-term body weight management in Chinese overweight adults or adults with obesity.

## Methods
### Trial oversight
The study was conducted in accordance with local laws, the International Conference on harmonisation Good Clinical Practice guidelines, and the ethical principles outlined in the Declaration of Helsinki. The study protocol and informed consent form were approved by ethics committees of all study sites: Peking University People's Hospital; The First Affiliated Hospital and Clinical Medicine College of Henan University of Science and Technology; The Fourth Affiliated Hospital of Harbin Medical University; Huzhou Central Hospital; Shandong Province Qianfoshan Hospital; Pingxiang People' s Hospital; The First Affiliated Hospital of Bengbu Medical College; Jinan Central Hospital; The Second Hospital of Hebei Medical University; Chu Hsien-I

Memorial Hospital; Jiangsu Province Hospital; Sun Yat-Sen Memorial Hospital; Shanghai Tenth People's Hospital of Tong Ji University; Qilu Hospital of Shandong University; Inner Mongolia Autonomous Region People's Hospital; Luoyang Central Hospital; Tonghua Central Hospital; Jingzhou Central Hospital; The First Affiliated Hospital of Nanyang Medical College. All participants provided written informed consent before study entry, between June 11, 2021 and Oct. 1, 2021.

### Study design
This randomised, double-blind, placebo-controlled phase 2 study was designed to evaluate the efficacy and safety of mazdutide in Chinese overweight adults or adults with obesity. The study included a 3-week screening period, a 2-week lead-in period, a 24-week treatment period and a 12-week off-treatment follow-up period (Fig. S1). The study consisted of two parts. The first part assessed the efficacy and safety of mazdutide dosed up to 6 mg in overweight participants or participants with obesity. The second part assessed the efficacy and safety of mazdutide 9 mg in participants with obesity (BMI ≥ 30 kg/m²). The enrolment, operation and data analysis of these two parts were independent. The first part has completed with a database lock on October 21, 2022 and unblinding procedure per protocol. This article reported the results of the interim analysis of the first part, which was approved by the ethics committee to guide the phase 3 study of mazdutide up to 6 mg in Chinese overweight adults or adults with obesity. The results of the second part will be reported in an independent publication.

## Participants

Participants were eligible for the study if they were adults (aged 18–75 years, both inclusive), overweight (BMI ≥ 24 kg/m²) accompanied by hyperphagia and/or at least one weight-related comorbidity (pre-diabetes, hypertension, dyslipidaemia or fatty liver within 6 months before screening; weight-bearing joint pain; obesity-related dyspnoea or obstructive sleep apnoea syndrome), or with obesity (BMI ≥ 28 kg/m²); had body weight change less than 5% during the 2-week lead-in period; were able to understand the procedures and methods of the study and willing to comply with the study protocol; and were willing to sign the informed consent form.

Participants were excluded if they had known allergies to the study drug or any components of the formulation or drugs of the same class; had self-reported body weight fluctuated by more than 5% in the 12 weeks before screening with diet and exercise alone; had used GLP-1 receptor agonists and co-agonists, anti-obesity drugs, glucose-lowering drugs, Chinese herbal medicine or health products that affect body weight, or investigational medication in other clinical trials within 3 months before screening; had a HbA1c level of 6.5% or above at screening or history of type 1 or type 2 diabetes; had a screening fasting blood glucose level of 7.0 mmol/L or above or 11.1 mmol/L or above 2 hours after a 75 g oral glucose tolerance test; had severe hypoglycaemia or recurrent symptomatic hypoglycaemia two times or more within 6 months; had obesity caused by increased cortisol hormone (such as Cushing's syndrome), pituitary gland or hypothalamus injury, or dose reduction or withdrawal of weight loss drugs; had a history of bariatric surgery or planned bariatric surgery or acupuncture for weight loss, liposuction and abdominal fat removal during the study; had a history of moderate to severe depression or severe mental illness; had previous suicidal ideation or suicidal behaviour; had PHQ-9 of 15 points or more at screening; Had answered "yes" to either Question 4 or Question 5 on the "Suicidal Ideation" portion of the C-SSRS at screening; had major and medium-sized surgery, severe trauma or severe infection within 1 month before screening and were determined by the investigator as not being suitable for this study; had blood donation and/or blood loss ≥400 mL or bone marrow donation within 3 months before screening, or hae-moglobinopathy, haemolytic anaemia, sickle cell anaemia.

Medical history and concomitant disease-related exclusion criteria included retinopathy; uncontrolled hypertension within 1 month before screening; malignancy; myocardial infarction, angina pectoris, acute and chronic heart failure, cardiomyopathy, or cardiac surgery; haemorrhagic or ischemic stroke or transient ischemic attack within 6 months before screening; thyroid C-cell carcinoma, multiple endocrine neoplasia 2A or 2B, or relevant family history; acute or chronic pancreatitis, gallbladder disease, and pancreatic injury; Heart rate <50 beats/min or >100 beats/min on 12-lead ECG at screening; clinically significant cardiac condition at screening including second or third degree atrioventricular block, long QT syndrome, or QTcF >500 ms, left or right bundle branch block, pre-excitation syndrome or other significant arrhythmia (except sinus arrhythmia); average weekly alcohol intake of more than 21 units for males or 14 units for females; and history of drug abuse.

Physical examination and laboratory results-related exclusion criteria included serum calcitonin ≥20 ng/L; ALT or AST > 3 × the ULN; serum total bilirubin (TBil) level >1.5 × the ULN; estimated glomerular filtration rate (eGFR) < 60 ml/min/1.73 m² at screening; fasting trigly-cerides >5.64 mmol/L; thyroid-stimulating hormone <0.4 mIU/L or >6.0 mIU/L; blood amylase or lipase >2.0 × the ULN; international normalised ratio (INR) of prothrombin time >the ULN; haemoglobin <110 g/L (males) or <100 g/L (females); and serological evidence of HBV, HCV, HIV and syphilis infection at screening.

## Randomisation and masking

A block randomisation method with a block size of 24 was used to randomly assign eligible participants 3:1:3:1:3:1 to receive mazdutide 3 mg, 4.5 mg, 6 mg or matching placebo using an interactive web response system, stratified by BMI at screening (<28 kg/m² vs ≥28 kg/m²). Randomisation list was generated by an independent statistician who was not involved in the clinical operations of the study. The study drugs and placebo were iden-tically labelled and indistinguishable in appearance. The partici-pants, investigators, study site personnel involved in treating and assessing participants and sponsor personnel in contact with the investigators and participants were masked to treatment alloca-tion until the database lock of the first part, while other sponsor personnel remained blinded until all participants in the first part had finished week 24.

## Procedures

Mazdutide and placebo were provided as once-weekly and sub-cutaneous injections in prefilled devices. For each mazdutide dose level, there was a matching placebo with the same injection volume and doing schedule. Mazdutide was administered with one of the three doses: 3 mg (1.5 mg weeks 1–4; 3 mg weeks 5–24), 4.5 mg (1.5 mg weeks 1–4; 3 mg weeks 5–8; 4.5 mg weeks 9–24) and 6 mg (2 mg weeks 1–4; 4 mg weeks 5–8; 6 mg weeks 9–24). The dose escalation schedules are shown in Fig. S1.

Participants were required to maintain the original diet, exercise and lifestyle during the trial. Study visits occurred at screening, before and after lead-in, day 1 (randomisation), every week through week 10, week 12, week 16, week 20, week 23, week 24 and every 4 weeks thereafter through week 36. Body weight, vital signs, electro-cardiogram and adverse events were monitored at every visit. Waist circumference was measured at screening, end of lead-in, day 1 and every 4 weeks thereafter through week 36. Local laboratory parameters were monitored at screening, day 1, every 4 weeks thereafter through week 24 and week 36. Central laboratory parameters (lipids and HbA1c) were monitored at day 1, week 12 and week 24 (Wuxi AppTec Inc., *Shanghai*). Pharmacodynamic (fasting plasma glucose and fasting insulin) and immunogenicity parameters were monitored at day 1 and weeks 2, 4, 8, 16, 24 and 32 (Wuxi AppTec Inc., *Shanghai*). Impact of Weight on Quality of Life–Lite (IWQOL-Lite) questionnaire assessment was conducted at baseline, week 12 and week 24. PHQ-9 and C-SSRS assessment were conducted at baseline and weeks 12, 24 and 36.

Participants who discontinued the study treatment before week 24 were requested to attend the remaining visits. Participants who discontinued the study before week 24 were requested to undergo end-of-study procedures as those who completed the study.

## Endpoints

The primary endpoint was percentage change from baseline to week 24 in body weight. Secondary efficacy endpoints included proportion of participants achieving body weight loss of 5% or more and 10% or more from baseline to week 24, change from baseline to week 24 in body weight, waist circumference and BMI, change from baseline to week 24 in glucose metabolism parameters (HbA1c, fasting plasma glucose and fasting insulin), changes from baseline to week 24 in cardiovascular risk factors (blood pressure, lipids [total cholesterol, LDL cholesterol, HDL cholesterol and triglycerides]), changes from baseline to week 12 and week 24 in serum uric acid and ALT, changes in IWQOL-Lite scores, as well as changes from baseline to week 36 in body weight, waist cir-cumference, BMI, glucose and cardio-metabolic risk factors.

Safety endpoints included safety, tolerability and immunogenicity of mazdutide. Safety was assessed by adverse events, vital signs, laboratory investigations and electrocardiogram. Treatment-emergent adverse events (TEAEs) were monitored from day 1 to the end of off-treatment follow-up and were categorised according to the Medical Dictionary for Regulatory Activities system organ class and preferred terms. The severity of the TEAEs (mild, moderate or severe) and the association between an event and the study treatment was assessed by

investigators based on pre-specified criteria. Hypoglycaemic episodes were identified by self-report or a plasma glucose concentration of 3.9 mmol/L or less at a site visit, and graded according to the American Diabetes Association criteria. Secondary safety endpoints also included C-SSRS and PHQ-9 assessments.

## Statistical analysis

The sample size of 240 participants was estimated to provide at least 95% power to show the superiority of optimal mazdutide dose to placebo in terms of body weight change from baseline to week 24, at a two-sided significance level of 0.05. In the calculation of the sample size, it was assumed the mean percentage change in body weight from baseline to week 24 of 2.3% with placebo and −10% with optimal mazdutide dose, a common standard deviation of approximately 8%, and a dropout rate of 20%.

The efficacy endpoints were evaluated in mITT population, defined as all participants who received at least one dose of the study treatment and had both baseline and post-baseline body weight measurements. The safety endpoints were evaluated in the safety population, defined as all participants who received at least one dose of the study treatment.

The primary analysis for the primary endpoint was done using ANCOVA model with treatment and stratification factor as fixed effects and the baseline body weight value as a covariate. Missing data at week 24 were imputed using last observation carried forward method. Treatment difference versus placebo were provided, with 95% CI and $p$ values. A sensitivity analysis of the primary endpoint using ANCOVA model as mentioned above except that missing data at week 24 were imputation using multiple imputation approach assuming missing at random. Another sensitivity analysis of the primary endpoint was done using MMRM, with treatment, visit, treatment-by-visit interaction and stratification factor as fixed effects, and baseline body weight as a covariate. An unstructured covariance matrix was used to model relationship of within-participant errors.

Continuous secondary efficacy endpoints were analysed using MMRM as the sensitivity analysis for the primary endpoint, with corresponding baseline values as covariates. For categorical secondary endpoints of body weight loss of 5% or more and 10% or more at week 24, Clopper-Pearson method was used for within-group CI calculation and Mantel-Haenszel method was used for between-group CI calculation. Cochran-Mantel-Haenszel method was used for statistical testing. Participants without week 24 body weight assessment were treated as non-responders. Post-hoc analysis of participant with body weight loss of 15% or more at week 24 was done using same method above.

Safety data were summarised descriptively.

All statistical analyses were done using SAS version 9.4. This study is registered with ClinicalTrials.gov, number NCT04904913.

## Reporting summary

Further information on research design is available in the Nature Portfolio Reporting Summary linked to this article.

# Data availability

Data underlying the analyses in this study cannot be publicly available due to the sponsor's (Innovent Biologics) contractual obligations and data privacy laws. Innovent Biologics will provide individual de-identified participant data underlying the results reported in this article. Data are available to request 6 months after the acceptance of this article. No expiration of data requests is currently set. Requests should be made to the corresponding authors (L.J. [jiln@bjmu.edu.cn] or L.Q. [cnradium@126.com]) and will be evaluated within 3 months. Access is provided after proposed use of the data has been approved by a review committee and receipt of a signed data access agreement with Innovent Biologics. The clinical study protocol and statistical analysis plan are provided as Supplementary Note 1 and Supplementary Note 2, respectively. Source data are provided with this paper.

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

## Acknowledgements

We thank all the participants, investigators and study site staff who were involved in the conduct of this trial. This study was sponsored by Inno-vent Biologics, Inc. The sponsor was involved in the study design, data collection, data review, data analysis, data interpretation and writing of the report.

## Author contributions

L.J., L.Q. and H.D. designed the study. L.J., H.J., Z.C., W.Q., L.Liao., Y.Z., X.L., S.P., L.Z., L.C., T.Y., Y.L. and S.Q. did the trial and collected the data. J.W., J.G., Y.W., L.Li. and Q.M. analysed the data. L.Q., H.D., J.W., J.G., Q.M. and H.H.-Z. interpreted the data. Q.M. and H.H.-Z. wrote the manuscript. L.J., L.Q. and L.Li. assessed and verified the data. All authors had full access to all the data in the study and had critically reviewed the manuscript and approved the final manuscript. All authors vouch for data accuracy and fidelity to the protocol.

## Competing interests

L.J., H.J., Z.C., W.Q., L.Liao., Y.Z., X.L., S.P., L.Z., L.C., T.Y., Y.L. and S.Q. received research funding from Innovent Biologics, Inc., during the conduct of the study. J.W., J.G., H.D., Y.W., L.Li., H.H.-Z., Q.M. and L.Q. were employees of Innovent Biologics, Inc.

## Additional information

**Peer review information** *Nature Communications* thanks Amanda Adler, Richard Pratley and the other, anonymous, reviewer(s) for their con-tribution to the peer review of this work. A peer review file is available.

[1]Department of Endocrinology and Metabolism, Peking University People's Hospital, Beijing, China. [2]The First Affiliated Hospital and Clinical Medicine College, Henan University of Science and Technology, Luoyang, China. [3]Department of Endocrinology and Metabolism, The Fourth Affiliated Hospital of Harbin Medical University, Harbin, China. [4]Department of Endocrinology, Huzhou Central Hospital, Huzhou, China. [5]Department of Endocrinology, Shandong Province Qianfoshan Hospital, Jinan, China. [6]Department of Endocrinology, Pingxiang People's Hospital, Pingxiang, China. [7]Department of Endocrinology, The First Affiliated Hospital of Bengbu Medical College, Bengbu, China. [8]Department of Endocrinology, Jinan Central Hospital, Jinan, China. [9]Department of Endocrinology, The Second Hospital of Hebei Medical University, Shijiazhuang, China. [10]NHC Key Laboratory of Hormones and Development, Tianjin Key Laboratory of Metabolic Diseases, Chu Hsien-I Memorial Hospital & Tianjin Institute of Endocrinology, Tianjin Medical University, Tianjin, China. [11]Department of Endocrinology, Jiangsu Province Hospital, Nanjing, China. [12]Department of Endocrinology, Sun Yat-Sen Memorial Hospital, Sun Yat-Sen University, Guangzhou, China. [13]Department of Endocrinology, Shanghai Tenth People's Hospital of Tong Ji University, Shanghai, China. [14]Innovent Biologics, Inc., Suzhou, China. [15]These authors contributed equally: Linong Ji, Hongwei Jiang, Zhifeng Cheng, Wei Qiu. ✉e-mail: jiln@bjmu.edu.cn; cnradium@126.com

