## [Peer Review File · Nature Communications]

A phase 2 randomised controlled trial of mazdutide in Chinese overweight adults or adults with obesityEditorial Note: This manuscript has been previously reviewed at another journal that is not operating a transparent peer review scheme. This document only contains reviewer comments and rebuttal letters for versions considered at *Nature Communications*.

REVIEWER COMMENTS

Reviewer #1 (Remarks to the Author):

This is a review of a revised manuscript of a randomised, double blind, placebo controlled phase 2 trial of mazdutide, a once weekly GLP-1 and glucagon receptor dual agonist in Chinese adults with overweight or obesity. Inclusion criteria were those aged 18-75 years, with BMI equal or greater than 24 with hyperphagia and/or at least one obesity related comorbidity, or obesity with BMI >28. The % body weight change from baseline was -6.7, -10.4 and -11.3 % for mazdutide 3mg, 4.5mg and 6mg respectively.

The authors have satisfactorily addressed the majority of reviewer's comments. The authors have now reported psychiatric endpoints, which suggest mazdutide is generally safe. Dose reductions occurred in a minority of patients on 4.5mg and 6mg doses and which suggest the study drug reasonably tolerated. The authors have appropriately addressed AEs lipase elevation and cardiostimulatory effects of the study drug in the discussion. The glucose tolerance data was also of interest and added to the manuscript.

The findings of this phase 2 study, alongside other trials of dual and triple GLP1-ra based agonists are important, though acknowledging this was in a single ethnicity.

I have the following comments

- The authors now clarify there are two parts – the first part assessed mazdutide dosed up to 6 mg in those with overweight and obesity (stratified by BMI above and below 28 kg/m²) and a second part assessed mazdutide 9mg in those with BMI ≥ 30k kg/m². Although allegedly the two parts are regarded as independent study, the authors should justify why the study was designed in such as way. Would it be more meaningful for the full results including the 9mg dose to be presented in one single publication? In actual fact, the mean BMI in the current part 1 study was 31.8 kg/m² and 84% had BMI over 28, only 16% in the overweight range. Is this weight/BMI much higher than planned for part 1 and implications for the choice of dose in the phase 3 study.

- Methods, eligibility criteria; the terms pituitary and hypothalamic injury remain ambiguous

Reviewer #3 (Remarks to the Author):

Thank you for addressing my comments.

Reviewer #5 (Remarks to the Author):

Professor Linog Ji and fairly straightforward P2 trial of a GLP-1/Glucagon coagonist in overweight/obese Chinese patients. The efficacy, safety and tolerability profile are as expected. The study was well designed, of high quality (> 90% retention) and the manuscript is a model of clarity.

Specific comments:

1) The DXA study of body composition is of small size, particularly with the treatment arms. The results as a consequence are not particularly compelling. Recommend deleting as this data is not robust.

REVIEWER COMMENTS

Reviewer #1 (Remarks to the Author):

This is a review of a revised manuscript of a randomised, double blind, placebo controlled phase 2 trial of mazdutide, a once weekly GLP-1 and glucagon receptor dual agonist in Chinese adults with overweight or obesity. Inclusion criteria were those aged 18-75 years, with BMI equal or greater than 24 with hyperphagia and/or at least one obesity related comorbidity, or obesity with BMI >28. The % body weight change from baseline was -6.7, -10.4 and -11.3 % for mazdutide 3mg, 4.5mg and 6mg respectively.

The authors have satisfactorily addressed the majority of reviewer's comments. The authors have now reported psychiatric endpoints, which suggest mazdutide is generally safe. Dose reductions occurred in a minority of patients on 4.5mg and 6mg doses and which suggest the study drug reasonably tolerated. The authors have appropriately addressed AEs lipase elevation and cardiostimulatory effects of the study drug in the discussion. The glucose tolerance data was also of interest and added to the manuscript.

The findings of this phase 2 study, alongside other trials of dual and triple GLP1-ra based agonists are important, though acknowledging this was in a single ethnicity.

I have the following comments

- The authors now clarify there are two parts – the first part assessed mazdutide dosed up to 6 mg in those with overweight and obesity (stratified by BMI above and below 28 kg/m²) and a second part assessed mazdutide 9mg in those with BMI ≥ 30 kg/m². Although allegedly the two parts are regarded as independent study, the authors should justify why the study was designed in such as way. Would it be more meaningful for the full results including the 9mg dose to be presented in one single publication? In actual fact, the mean BMI in the current part 1 study was 31.8 kg/m² and 84% had BMI over 28, only 16% in the overweight range. Is this weight/BMI much higher than planned for part 1 and implications for the choice of dose in the phase 3 study.

Response: *We thank the reviewer for the comments. When the study was launched, the non-clinical toxicity study to support mazdutide 9 mg was not ready. The addition of 9 mg cohort as an amendment in this phase 2 study maximized the flexibility of clinical operation to expedite the overall clinical development by Innovent Biologics.*

*As we declared in the protocol, the enrolment, operation and data analysis of these two parts were mutually independent. The enrolment of the second part began after the double-blind treatment period of the low-dose cohorts. The first part has completed with a database lock on October 21, 2022 and unblinding procedure per protocol. The participants, investigators, study site personnel involved in treating and assessing participants and sponsor personnel in contact with the investigators and participants were masked to treatment allocation until the database lock of the first part, while other sponsor personnel remained blinded until all participants in the first part had finished week 24. We added the above details in study design (**Line 109**) and randomisation and masking (**Line 168**) section of Methods.*

To ensure timely disclosure of the results of the low-dose cohorts, we chose to publish the results of

low-dose and high-dose cohorts separately. We revised the abstract to state that this report of mazdutide up to 6 mg in Chinese adults with overweight or obesity served as “an interim analysis of a randomized, two-part (low doses up to 6 mg and high dose of 9 mg), double-blind, placebo-controlled phase 2 trial” (Line 45). We also stated in the Methods that this interim analysis of the first part was approved by the ethics committee to guide the phase 3 study of mazdutide up to 6 mg in Chinese adults with overweight or obesity (Line 110).

We did not pre-specify the proportion of the participants with overweight in the low-dose cohorts and the participants were recruited randomly. The observed proportion roughly reflected the distribution of BMI of participants in the clinics. The large proportion of participants with baseline BMI over 28 kg/m² and the flattening curve of body weight reduction at week 24 (Fig. 2b) implied that participants with higher BMI may need higher dose to achieve desirable weight loss targets. This was one of the reasons that mazdutide 9 mg cohort was launched. Moreover, mazdutide 4 mg and 6 mg were under development for the treatment of type 2 diabetes. Thus, mazdutide 4 mg and 6 mg may be indicated for the treatment of overweight and mild obesity, with or without type 2 diabetes, while mazdutide 9 mg may be indicated for the treatment of moderate to severe obesity.

- Methods, eligibility criteria; the terms pituitary and hypothalamic injury remain ambiguous

Response: We thank the reviewer for the comments. The criterion was set to exclude participants with a history of acquired hypothalamic obesity caused by hypothalamic damage that disrupted the hypothalamic-pituitary axis, and those with obesity and hyperphagia after traumatic brain injury that impacted the production of pituitary hormones.

Reviewer #5 (Remarks to the Author):

Professor Linong Ji and fairly straightforward P2 trial of a GLP-1/Glucagon coagonist in overweight/obese Chinese patients. The efficacy, safety and tolerability profile are as expected. The study was well designed, of high quality (> 90% retention) and the manuscript is a model of clarity.

Specific comments:

1) The DXA study of body composition is of small size, particularly with the treatment arms. The results as a consequence are not particularly compelling. Recommend deleting as this data is not robust.

Response: We thank the reviewer for the comments. We agreed with the reviewer that the sample size of DEXA was small, due to COVID-19 pandemic-related restrictions during the conduct of the trial. Endpoints regarding body composition were exploratory and the results were preliminary. We acknowledged this limitation in the Discussion. Nevertheless, these limited data provided the first glimpse of the effect of mazdutide on body composition. To the best of our knowledge, mazdutide was the first GLP-1 and glucagon receptor dual agonist that disclosed the effect on body composition, arguing against the potential lean mass loss by catabolic effect of glucagon.

REVIEWERS' COMMENTS

Reviewer #1 (Remarks to the Author):

Thank you for the detailed responses which have further improved the clarity of the manuscript. I have no further comments.